# "It Really Put a Change on Me": Visualizing (Dis)connections within a Photovoice Project in Peterborough/ Nogojiwanong, Ontario

**Rosa Lea McBee**

Department of Sustainability Studies, Trent University, Peterborough, ON K9L 0G2, Canada; rlmcbee@uvic.ca

**Abstract:** Photovoice is an arts-based participatory action research method that uses photography as a means for individuals, usually those facing marginalization, to document and foster group dialogue around the stories of their valuable lived experiences. This paper details a photovoice project run under the participatory planning project NeighbourPLAN, in Peterborough, Ontario, with the residents of the Downtown Jackson Creek group. The focus of the photovoice project was working with residents facing various forms of marginalization and barriers to reflect on what (dis)connections look like in their community. The findings conclude that photovoice generated new subjectivities, as residents reported feeling more connected to their community after taking photos. The process was generative in that it reminded residents of other creative outlets that they enjoyed doing and inspired them to engage with creative reflection in other ways. The findings also determined that green spaces, non-judgmental institutions, accessible amenities, safe housing, and well-maintained streets were critical for resident researchers' feelings of connectedness. I conclude with recommendations from the residents' feedback on the method and project, along with highlighting the promising potential of arts-based and storytelling methods when conducting research with marginalized groups.

**Keywords:** photovoice; arts-based methods; social connectedness; participatory planning; community engagement

## 1. Introduction

With growing concern around economic, social, and environmental inequities affecting communities across Canada, participatory engagement methods have become a growing practice in academia, health initiatives, city planning, and beyond. Participatory action research (PAR) principles are commonly drawn upon to examine the link between public health and urban development, especially among vulnerable groups (Arcaya et al. 2018). PAR reframes the traditionally silenced "subjects" of research as key "knowledge producers", aware of their experiences, capable of self-representation, and entitled to mutual benefit that should be generated during and after any research process (Askins and Pain 2011). Concurrently, a growing awareness of the health impacts of loneliness (Griffiths et al. 2007; Hari 2018) and subsequent desire to heal the atrophying social spheres and sense of connectedness in communities, guided this research paper to explore the potential of PAR to address deteriorating social trust and engagement. *Photovoice* is a qualitative, arts-based participatory research method first introduced and implemented by researchers Wang and Burris (1997), who draw inspiration from Paolo Freire's concept of *conscientização* or critical consciousness-raising. Freire's critical pedagogy often started with community members drawing or photographing their community to encourage reflections on the social and political systems that negatively impact them (Freire 1970). Photovoice realizes the spirit of PAR as it invites participants to identify, collect, and analyze their world, realities, and experiences by taking photographs. The photographs encourage creative and visual representations of assets and challenges in a community, which is accompanied by critical

dialogue through focus groups, or public exhibitions around selected images (Hergenrather et al. 2009; Joyce 2018; Wang and Burris 1997).

This article details a photovoice project conducted under the umbrella of a multi-year participatory planning project, NeighbourPLAN, while exploring relationship-building within PAR activities, and how building social connectedness can be intentionally fostered within research. Beauregard et al. (2020) define connectedness as the quality and number of ties individuals maintain with others, embracing "areas such as family, peers, school, and community" that "promotes well-being, increases adaptive capacity, and enhances a sense of belonging" (p. 438). Community engagement is becoming central to planning on a local level, and storytelling is more important than ever (Walljasper 2018, p. 29); thus, not-for-profits, researchers, community organizers, and planners are increasingly adopting PAR methodologies when working with local citizens or "lay experts" to create more equitable involvement and reciprocal outcomes (Arcaya et al. 2018). The focus of the photovoice project was to create opportunities for storytelling and reflections on connectedness within a larger participatory project by visually capturing what (dis)connections look like in a low-income neighbourhood facing various forms of marginalization. The findings of this project show that through the process of documentation and photo-storytelling, residents shifted their experience of their community to feel *more* connected than before and clarified the spaces and places that offered safety and those that did not. Photovoice, particularly in participatory planning initiatives, excels at giving narrative support to intangible feelings and experiences for more textured, nuanced, and complex meaning to emerge that is generative of new possible understandings for participants and facilitators. The photovoice project made participants' worlds within Peterborough more visible to themselves, to one another, and to the other stakeholders in the larger NeighbourPLAN project.

## 2. Background

GreenUP, a Peterborough/Nogojiwanong, Ontario-based non-profit focused on urban environmental sustainability, launched the three-year participatory planning project 'NeighbourPLAN' in 2017, with the vision that there is a mutual benefit when residents are included as planning partners (Nasca 2016). NeighbourPLAN worked with 3 neighbourhoods, 18 organizations, 2 Universities, and numerous city councillors, planners, and architects as they sought to be a third-party broker between city planning and underrepresented, marginalized neighbourhoods. Macedo (2000) warns against using euphemisms like "marginalized" or "disenfranchised" instead of "oppressed" because it risks making the Oppressor invisible. While we can and should expose oppressive institutions and those perpetuating them, I am uncomfortable referring to the people I met through this NeighbourPLAN as "oppressed" as it does not make apparent the agency, skills, power, and creativity of the people I spoke with. For this reason, I will use the term "marginalized" to refer to any group that lives outside the white cisgender, heteronormative, able-bodied, financially secure norm, understanding that this phrase is insufficient and risks obfuscating the severity of many realities.

Within the initial surveys within the Downtown Jackson Creek (DTJC), residents wished for safe, accessible spaces where "residents are able to make new relationships and feel connected to one another through participating in the project (GreenUP Association and The Centre for Active Transportation 2019, p. 6). As a graduate student recruited into NeighbourPLAN to conduct embedded research and an evaluation of the program, this anticipated outcome piqued my curiosity about the social ties and networks formed through participatory processes. Since NeighbourPLAN primarily sought feedback from residents through focus groups, pop-ups, and surveys, I was intrigued by what residents would think about an arts-based methodology. I used photovoice to address how NeighbourPLAN participants would define social connectedness and what connections look like in their communities. I use "connectedness" when referring to relationships, or the social 'glue' in a community, rather than the more prevalent term "social capital", coined by economist Glenn Lour, as the latter is often co-opted by neoliberal discourse (Small 2009). This

discourse can assume that if people living in poverty networked better and made "good" 'investments' through relationships, their social circumstances would improve, bypassing speaking to systemic issues such as class relations or the state's role in developing or inhibiting connectedness in communities. (Mohan and Stokke 2000).

### 2.1. Downtown Jackson Creek (DTJC)

While NeighbourPLAN worked in three neighbourhoods across Peterborough, this paper focuses on the photovoice project I conducted with the DTJC neighbourhood. DTJC, a highly populated region west of downtown Peterborough, is a diverse, low-income neighbourhood with a higher-than-average unemployment rate of 12.7% compared to 8.9% citywide (GreenUP Association and The Centre for Active Transportation 2019). Within the same report, the 2019 average household income was shown to be 34,058 CAD, and 86.6% of residents are tenants, compared to the 37.9% tenancy average in the rest of Peterborough.

NeighbourPLAN planned three phases in each neighbourhood: the portrait phase, engaging residents to determine concerns and assets; the vision phase, to set out goals and priorities (see supplementary materials); and the action phase, to GreenUP's network to advocate for the ideas generated by residents (Active Neighbourhoods Canada 2015). While NeighbourPLAN supported residents with knowledge creation and mobilization, it could not guarantee any of the project's ideas. It is the city that ultimately decides whether design proposals are 'feasible' or not, leaving much uncertainty toward the project's end, reminding us that "sharing through participation does not necessarily entail sharing in power" (White 1996, p. 143).

### 2.2. Intentional Connection Building in PAR

The interdisciplinary research and evidence of social connectedness as an important factor for health and wellness is extensive, and factors such as individual and community resilience, diagnoses, and recovery all improve with the presence of meaningful connections (Comes 2016; Griffiths et al. 2007; Hari 2018; Taylor and Wei 2020; Umberson and Montez 2010). According to Small (2009), to examine and minimize inequality in well-being due to network inequities, we must first understand how people's social connections are formed. Despite substantial evidence backing its perceived importance, many academic disciplines take social bonds and the mechanisms that establish them for granted (Mayan and Daum 2016; Small 2009). Even the three most notable and prolific writers on Social Capital, Bourdieu, Lin, and Coleman (in Small 2009), sidestepped the subject of how to develop connections. Within this gap, connection-building is glossed over as serendipitous or dependent on the facilitator's leadership and charisma. Of course, practicing reciprocity, promoting listening and learning, and ensuring continual interactions are important steps to creating the foundation for relationships (Hall and Tandon 2017; Hardy et al. 2018; Kesby 2005; Levkoe and Kepkiewicz 2020); however, there is not much documentation of the intentional decisions behind developing connections through the research process.

Participatory techniques already have the potential to be relational, requiring cooperation and diverse stakeholders to work together, who significantly benefit from existing social networks (Janzen et al. 2016; Kemmis and McTaggart 2005). Levkoe and Kepkiewicz (2020) performed a pan-Canadian evaluation of 12 community engagement initiatives and concluded that after participatory research projects, partners perceived other partnering organizations as better colleagues and friends working towards a common goal and agreed that the process broadened networks, leading to future collaborative works.

However, relationships and connections are difficult to anticipate, explain, and evaluate, which can come at direct odds with the bias among funding agencies' criteria to evaluate investments by citing measurable, observable, and quantifiable change, which is equated to impact in final reports. (Hardy et al. 2018; Levkoe and Kepkiewicz 2020). Hardy et al.'s (2018) case study of Hermosas Vidas is unique in that it seeks to quantify the number of 'lasting' relationships built through the participatory health research project. While evaluating the relationship building of the project by taking a cross-section of the stakeholders,

from the families to partnering institutions, they found that the institutional partners of the project reported creating up to five times as many new connections as the project's intended beneficiaries and most vulnerable stakeholders, the families dealing with health issues. The authors argue that this asymmetrical benefit, even if unintentional, contradicts PAR values of benefiting the community first. To explain this disparity, they created a Ripple-Effect Theory Model, which sets up PAR researchers with a framework to anticipate a gap in benefits, conduct cross-sectional evaluations, and adjust accordingly, mid-project.

*2.3. Prioritizing Connections through Arts-Based Storytelling*

Researchers often fear that they may inadvertently choose a method that causes more harm to vulnerable groups they seek to support. In their evaluation of a national participatory initiative in food security conducted in food banks, Bay and Swacha (2020) found that an electronic survey alienated and angered participants who already felt vulnerable and unseen while waiting in line for food services. The authors admitted how the priority of more data too quickly became extractive and dehumanizing and argue that as researchers, we must intentionally map out a process that encourages people's entanglement and connections, with methods that can also allow for the rich experience of daily life to be seen and heard. Cahill (2007) notes that participatory research is emotional listening and reflects on the therapeutic quality of PAR when sharing personal stories in a "collective space for breaking the silence" (p. 281). Expressing emotions through personal stories helps participants work through the confusion of systemic oppression and create solidarity among the group. This is where arts-based storytelling methods, such as photovoice, find their fit.

The literature about participatory research discusses connectivity and quintessential social networks but does not often explain the decisions on how to develop them. Encouragingly, participatory practitioners discuss this gap using narrative and arts-based strategies that disclose their creative processes and observe social relationships as an outcome. Arts-based and storytelling methods do not sidestep PAR's challenges with power dynamics, co-option, and resource constraints, but they offer a way to circumvent social norms as they demand unique dialogue, listening, and cooperation. They are one mechanism we can turn to when strengthening social networks and a sense of belonging in our communities is a priority.

**3. My Research Approach**

Despite the claim in the participatory literature that building relationships within participatory projects is essential and leads to more action (Janzen et al. 2016), there is a gap in the participatory literature on how connections and relationships are defined and how they are/were formed. NeighbourPLAN's 'Theory of Change[1]' reasoned that more connected residents would have more collective power to influence the services and spaces that mattered to them. Trying to quantify or qualify relationships is difficult due to their "deeply personal and emotional nature" (Levkoe and Kepkiewicz 2020, p. 233). However, I still wanted to ask, what made residents from the DTJC feel more connected to their community, and what disconnected them?

As part of my NeighbourPLAN evaluation role, I acted as an embedded researcher (Lewis and Russell 2011), participant, and member of the Evaluation Committee. I observed, participated in, discussed, collaborated on, and supported different project tasks, including attending resident meetings, for ten months before initiating the photovoice project with the DTJC resident group.

While I used two focus group evaluations with two neighbourhoods in the NeighbourPLAN project as my primary method to evaluate NeighbourPLAN's activities and to determine a context-specific definition of connectedness for residents, in this paper, I focus on the photovoice project I facilitated in the summer of 2019 with the DTJC neighbourhood group. I refer to all the photovoice residents by pseudonyms, which carries some tension around representation and credit. However, I kept them anonymous as many residents were also in my focus groups and giving critical feedback about the program. There were

five women and one man, 30–50 years of age, white, low-income, and three were known to be Ontario Disability Support Program recipients and living with chronic health or disabilities that impact how their bodies move through space.

## 4. Methodology

In traditional academic frameworks, lived experiences and local knowledge are difficult to translate; however, the methods are "key to deconstructing dominant discourses and social hierarchies" and asking a community to articulate its knowledge with conventional methods risks generating conventional answers (Askins and Pain 2011, p. 806; White 1996). In their article, Bay and Swacha (2020) warn participatory practitioners against weighing the institutional demands for quantifiable data over the value and necessity of slower, engaged, and sensitive research methods when interacting with vulnerable populations. After hearing those critiques, I chose a qualitative research design that would prioritize subjective and intersectional knowledge and allow for deeper experiential understandings (Given 2008). As a form of PAR, photovoice would provide a way to engage with the research process that, hopefully, would feel generative and allow residents to reflect differently than they had done previously through regular meetings, focus groups, events, and surveys. As the creators of photovoice, Wang and Burris (1997) outlined three main goals of the research method: (1) to empower research participants to be the ones to document and cogitate on their lives and circumstances, (2) to encourage critical dialogue and knowledge creation through the photographs with the group, and (3) to mobilize the photos and knowledge created to end up with policymakers.

Residents were recruited based on their interest in NeighbourPLAN, and GreenUP staff contacted the participants first. From there, snowball sampling (peer selection) recruited three more from the initial three participants (I had met with the three over the various meetings and events for the previous eight months). This photovoice project had three phases: an introductory workshop, two weeks to take photos, and a final discussion group where residents shared what they felt, learned, and wanted others to see in their photos. During the introductory workshop, I explained the method, gave quick photography tips such as the rule of thirds, lighting, etc., and distributed disposable cameras and printed out photo journals that prompted an examination of feelings, motivations, and desires. Given the two-week timeframe, I simplified Wang's (1999) popular photo-journaling technique, SHOwED[2] in the photo journals. I then asked residents to photograph and reflect on their neighbourhood's connections and disconnections without any other parameters to avoid limiting the possible manifestations, definitions, and perspectives. For example, despite the recommendations from Kindon et al. (2007) to identify a target audience, such as policymakers or service providers, I did not explicitly state a desired audience. Instead, I asked residents to think about whom they wanted to view their photos and to keep them in mind while they took photos and reflected in their journals. This flexibility led to some interesting outcomes but also some limitations, which I discuss further in the limitations section (see Section 4.1). Finally, I led a 90-min focus group where residents shared their photos, gave feedback on the method itself and then went into storytelling around the photos and used coloured stickers to start us thinking about how thematic analysis works. I iteratively returned to those themes, the transcript, photos, and photo journals, until all the data were categorized.

To be clear, while I evaluated a participatory planning project and conducted a literature review on PAR and arts-based participatory methods, I am not suggesting that the research design of this photovoice project strictly adheres to or meets the criteria of what I believe to be PAR. While photovoice allowed the residents to conduct their own data collection and photo journaling encouraged preliminary analysis, a lack of resources coupled with a concern of over-saturation limited us from being able to spend time together in the planning and analysis phases of research. During the photovoice focus group, residents placed coloured dots on photos based on loose themes, contributing to the final analysis. However, I conducted months of analysis work after that, thus privileging my meaning-

making over the residents' (Dassah et al. 2017). Regarding the design of the project, I initially suggested to the residents that they take photos within the neighbourhood boundaries set by NeighbourPLAN; however, residents expressed that those boundaries did not align with their concept of their community, which was an issue reiterated later in the focus groups that evaluated NeighbourPLAN as a whole. Expanding those boundaries to where they visit daily would better show their connections, which we agreed on. GreenUP was concerned that I not put too many demands on residents on top of all the events and activities that NeighbourPLAN was requesting of the residents, so it was always a negotiation of what should be prioritized. Indeed, GreenUP's concern was affirmed when I later conducted a focus group evaluating NeighbourPLAN and residents noted the burden of the work.

The final phase of photovoice typically showcases photos to inform policymakers or the community (Wang 1999; Joyce 2018). Time constraints did not allow for a public show, so the photos were used in the final DTJC Vision document, a public document that ended up on the desks of numerous partnering organizations, city planners, city officials, and more. The group also expressed interest in having their photos in local publications, *The River*, a magazine dedicated to showcasing the creativity and ideas of the marginalized and low-income in Peterborough. After submitting one self-selected photo and journal entry on behalf of each resident, the team at *The River* accepted three entries.

### 4.1. Limitations with Photovoice and the Research Design

High-quality photovoice projects can be costly, but with an internal grant from Symons Trust and some funds made available by GreenUP, I designed this project with honoraria (100 CAD per resident), cameras, developing film, and refreshments totalling approximately 1000 CAD. When asking for feedback and negotiations of the introductory workshop, I decided against a request from one resident to use personal cameras. I felt that using disposable cameras had benefits, such as forcing us to plan out film usage and ensuring that everyone's photos were the same quality since some residents did not have access to cameras. Several residents were inconvenienced by the technical issues of the disposable cameras, especially those with smartphones or better-quality cameras. I am still uncertain whether my attempt at creating equal conditions among participants outweighed the costs to the project. My hope was no one would feel embarrassment or exclusion due to a lack of resources, but by denying some residents their request to use their personal cameras, I potentially communicated to those residents that they were consultants rather than co-creators of the research design (Liebenberg 2022).

As discussed, I did not specify a target audience to participants which resulted in some interesting insights and drawbacks. While many photos were directed to the broader public or to me as the researcher, one resident addressed many of her photos and journal responses towards her fellow neighbours, inviting them to see the community's strengths by becoming more involved in the events and nearby parks and attractions. This resident identified her neighbours as the key to bringing about the change she wanted. I view this variability in the audience as a generative outcome, as residents had different concepts of where power and "agents of social change" were located, i.e., on either the political or community level (Carlson et al. 2006, p. 849). However, it also potentially created some disadvantages by lacking focus and targetedness when considering the explicit appeal that the methodology has to speak to powerful others to effect change (Liebenberg 2018).

While the photovoice project attracted socioeconomically diverse groups with different abilities who were already part of NeighbourPLAN's more consistent partners, neither NeighbourPLAN nor I were able to engage people of colour and Indigenous peoples in a lasting way, despite their presence in the downtown core. Exploring the absences and rejections surrounding every participation project is a chance to learn, but few research papers account for it. White (1996) says that non-participation can be as empowering as involvement for underprivileged groups if they feel the project will not benefit them. This highlights the limitation of what we offer through participation and how people may feel excluded based on ethnicity. Projects can improve this without abandoning participatory

models by directly addressing power, class, race, and gender (Cooke and Kothari 2001; Kesby 2005). Cahill (2007) suggests conceptualizing participatory spaces as "contact zones" that do not erase participants' social, economic, and cultural context while focusing on our tensions, exchanges, and reciprocity and by including more diverse representation within the organizing team (Torre 2010; Murdoch et al. 2016). I do not visibly represent that diversity, and while NeighbourPLAN had diverse gender, sexuality, age, and ability in the organizing team and core groups, they also lacked ethnic diversity.

Finally, half of the photovoice participants identified as living with a disability, but I did not use a disability arts framework for this photovoice project. I did not specifically recruit on the topic of disability; however, considering the themes of accessibility, stigma, and healthcare that emerged throughout the photos, and noting that a tremendous body of literature already exists precisely at the nexus of photovoice, disability advocacy, and rewriting harmful narratives around disability (Newman et al. 2009; Oden et al. 2010), I now see the oversight of not utilizing that framework. Especially as I was working with the DTJC neighbourhood group, where disability and low-income status are intertwined and constantly affecting each other.

## 5. Findings

During the debrief with residents, the discussion ranged from the personal to the systemic. The group seemed to enjoy what Bay and Swacha (2020) call 'affective time', listening and learning about each other and relating to what they heard, from favourite spots in town to negative experiences in institutions. Residents were optimistic about the project when I asked how the photovoice compared to the other participatory methods employed by NeighbourPLAN, responding that photovoice was "more interactive" and "more personal," and Patty added, "it's good to see your connections, see other people's [connections]". Residents evaluated their neighbourhood connections, prompted by the photos, and shared which judgment-free spaces supported their well-being. Furthermore, but not expectedly from a Frerian-based process, residents' collective reflections and conclusions quickly revealed the broader power deficits they feel negatively impact them, such as inaccessible housing and experiencing stigma in healthcare.

During the second half of the focus group, I asked participants to use stickers to organize their photos into themes and describe the stories they wanted to share through them. The preliminary categories set during this time were green spaces, leisure and activities, housing, heritage buildings and art, accessibility, unsafe spaces, health and wellness, and the places and people that made them feel they belonged. I allotted a 90-min session for this activity because some of the residents have chronic health issues and we needed to consider the discomfort sitting for long periods can cause. Additionally, sharing knowledge can be an intimate and emotionally exhausting process, so I consolidated some of the outlier categories into other existing themes. While having participants co-create the categories would have been preferable, I was aware of the time asks of participants within the greater project. Going forward, I would recommend that time and resources for this kind of co-creation be incorporated into future projects. Categorizing the photos into five categories: 'institutions and organizations', 'green spaces and leisure', 'accessibility', 'shifting subjectivities', and 're-igniting creative outlets', the paragraphs below synthesize conversations during the debrief and excerpts from the photo journals, accompanied by select representative photos.

### 5.1. (Dis)connections: Institutions and Organizations

The first observation in this category was that residents focused on places they felt connected to during the debrief. They were quick to share resources of people and places that made them feel safe and seen. They took many photos of essential local businesses, services, and institutions, listing the new Public Library, GreenUP, Good Neighbours Care Centre, urban gardening groups, and St Vincent de Paul food bank as "mainstays" with friendly, nonjudgmental staff and individuals. Residents felt connected to churches and historic

buildings (see Figure 1), which they thought should be more used and respected. Marie, a resident with mobility impairment, took photos of disappearing amenities near her home, making her feel disconnected because she now had to travel outside her neighbourhood to access those same services.

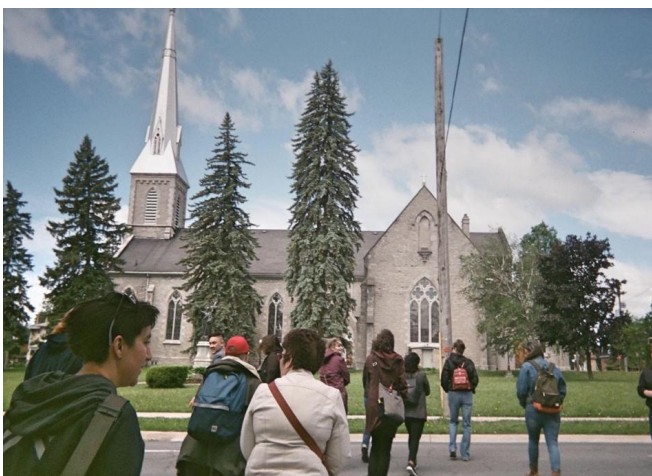

**Figure 1.** "Trinity Church—Church Row". Marie's entry: This is one of the few churches left still functioning on the row. I am seeing more and more history disappear, and it's disturbing. This church can be used for many other purposes than a place of worship. Let's keep these places of beautiful history running and keep the people going to them. Image Description: Church with tall steeple in background with a group of people crossing the street, walking towards the church.

I noted in the debrief that most participants had photographed the local hospital (see Figure 2). When I asked if this connection was positive or negative, they responded, "negative!" They then shared stories and strategies when being denied care due to stigma and racism. Jessica remarked, "They racially profile", and Patty noted their need to accompany a friend who is Indigenous to the hospital, to which Jessica and Marie agreed. Daniel shared that he had been neglected when visiting the emergency room multiple times in pain for what turned out to be cancer; "that's my problem—they see me and think 'oh, he's a junkie.' They see no teeth, ripped jeans, long hair—but that's just how I look". Through the discussion and journals, it was clear that these residents feel a deep sense of loss in this neighbourhood, taking photos of the hospital and spots where they made memories or visited with friends and family who have since passed.

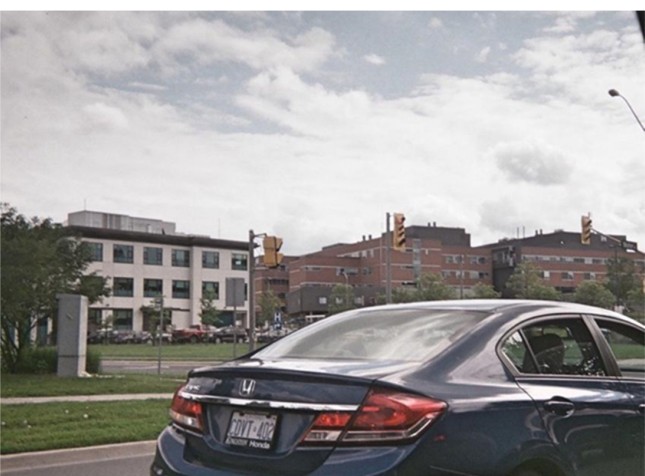

**Figure 2.** "Peterborough General Hospital". Patty's entry: This is the local hospital. I've taken the pic from a distance because I don't really want to be there. I have been in this hospital for myself and

many friends. Many are not here now. Some are still fighting for life. And I'm in that fight with her/my friend. Don't let anyone die alone. Say what you need to say now. Do what you need to do. There might not be a tomorrow. Image Description: Cluster of large hospital building taken from a car window with a parked car in the lower right corner of the image.

### 5.2. Green Spaces and Leisure

Green spaces and leisure was a ubiquitous theme within residents' photos for what helped them feel connected. Peterborough's green spaces, trails, urban gardens, rivers, and creeks are plentiful, and residents spoke of their appreciation for beautiful areas to travel to in the city, escape, fish, play, and visit with family (see Figure 3).

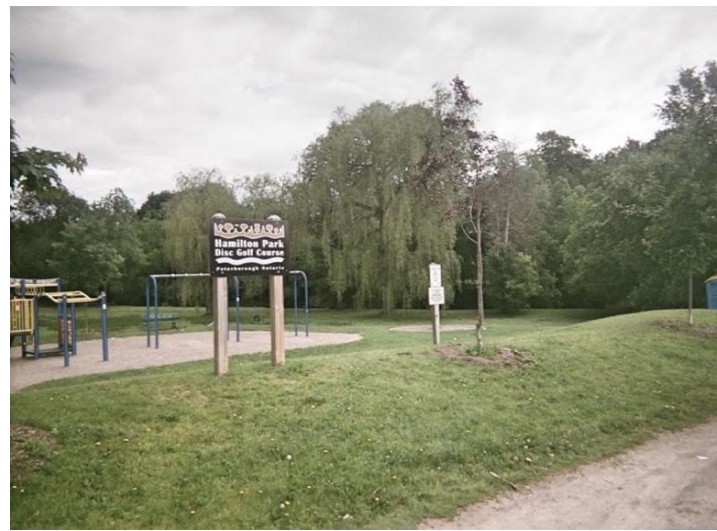

**Figure 3.** "Hamilton Park". Daniel's entry: Children enjoying themselves when school's out. I love taking my grandchildren. It is a place of happiness, and you leave with a smile. Image Description: A park with a playground in the background. A wooden sign reads "Hamilton Park".

### 5.3. (Dis)connections: Accessibility

Accessibility and Peterborough's transit system were common themes for residents with disabilities. As demonstrated by Marie's photo (see Figure 4), residents with mobility impairments felt disconnected from neglected areas and streets without sidewalks. Accessible parking spots at the Public Library and drugstores, as well as transit, make the residents with mobility devices and impairments feel more connected as they are able to navigate those spaces they have to frequent regularly. Patty took photos of the bus depot and wrote, "many people use this transit system to stay connected".

Housing accessibility is another critical issue for residents with mobility impairments. Patty took photos of her housing to show all the accessibility ramps into their building, noting it was the only place she could live with her mobility issues, but also the discord she felt, feeling safe having a community but also feeling very restricted and unsafe in the small basement space (see Figure 5). Concern was expressed that residents of diversity groups living in the building were vulnerable to crimes against them.

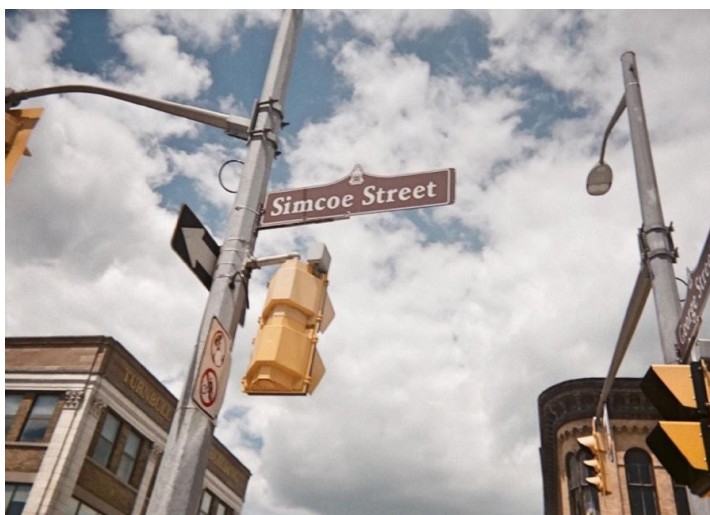

**Figure 4.** "Sheer Hell". Marie's entry—One of the worst streets in Peterborough for potholes, sinkholes, bad sidewalks. It is horrible for anyone with assisted devices. I use a walker and find it almost impossible to navigate the sidewalks. I want others to become aware of how hard it is to get around on some streets that are not kept up when you have an assisted medical device, not just a vehicle! Image Description: A street sign with a traffic light below it, blue skies, and clouds in the background.

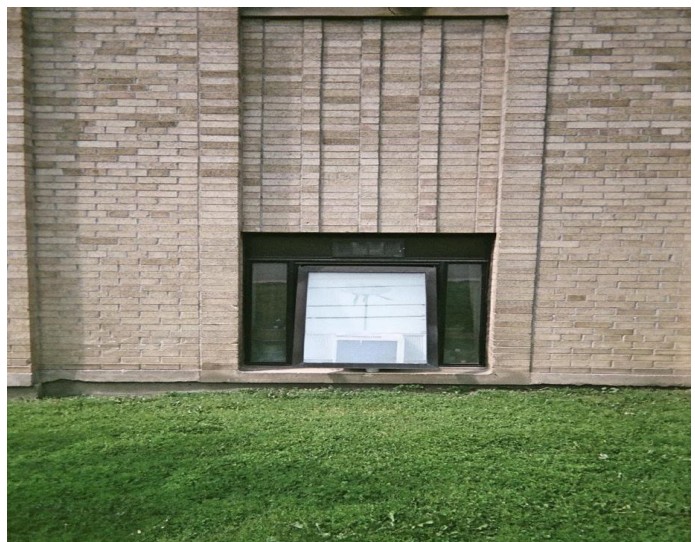

**Figure 5.** "My only window, ground-floor apartment". Patty's entry—Sometimes, I find this window unsafe. People have knocked and looked in the window, which is frightening. But I need a window so I can see the outside, the weather, the plants, the birds/animals. I need to connect to the outside. Basements are not really healthy apartments to live in! I stay in this apartment and this building because if I didn't and when I become more ill, I will need help from KPP 24-h helpline in the building. Unfortunately, living here is hard. Many of my friends have died. They lived here because they had precarious lives where they needed help. Image Description: Grass leading up to a brick building with one window just inches above the ground, opened from the bottom.

## 5.4. Shifting Subjectivities

As noted previously, PAR's concept of empowerment is quite vague. Cahill (2007) swaps the word "empowerment" for "shifting subjectivity" to mean redefining marginalized selfhood in a more critical or positive light. Residents in the photovoice project emulated this by describing profound changes in their perceptions of their neighbourhood when they started to think about their connections through a camera lens. Sarah asked herself, "what is my community?" continuing, "I really felt that sense of community. It

gave me great pleasure; it really intensified that feeling; it was great!" Marie continued after her, "I always thought where I lived was the building, and I hated it. But no, I live in the community, and I respect it. So, it really put a change on me". Patty said she felt more connected to her community after the activity. Sarah shared a story about an at-risk man in the neighbourhood of whom she took a photo. She shared that she initially was a bit wary of him, but after he shared his story, she saw how "scared and lonely" he was. He has become one of those familiar faces she looks forward to seeing around and fosters a sense of connectedness for her. She observed, "you sometimes connect with those people without even knowing about it. Approaching them with a camera is one way to start talking to someone".

### 5.5. Reigniting Creative Outlets

Four out of six residents said the process inspired them to reengage with their creative outlets (see Figure 6). Most had not taken photos in a while and enjoyed doing so again. Three residents said the project's photo journal was important; Sarah shared, "I have started journaling again! I used to do it and forgot how much I enjoyed it". Marie said, "I used to make a lot of scrapbooking, and this got me thinking about that again". Patty reflected, "I like to use being creative as therapy—to work out my emotions, my thoughts, and my needs".

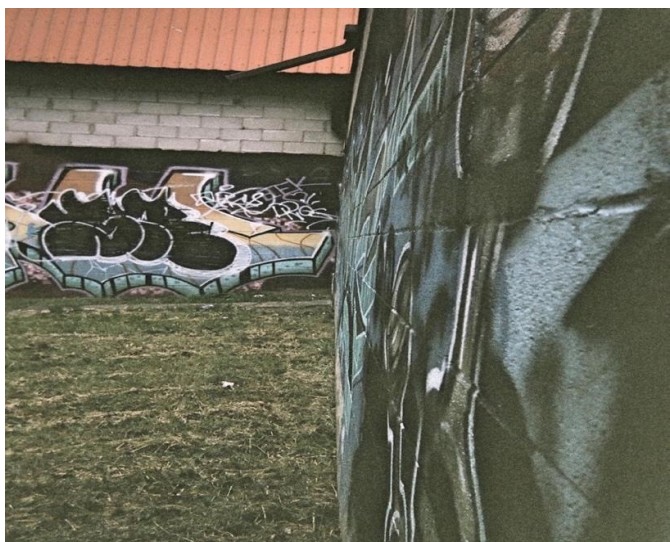

**Figure 6.** "Graffiti and Vibrant Colors". Sarah's Entry—I walk by so often but never appreciated it up close to really see the writing on the wall. That graffiti is part of a sociological statement. Different painters are combining different thematics, some angry, some playful. I always wanted to be an artist. Image Description: A graffiti wall up close, with another graffiti wall in the background.

### 5.6. Resident Feedback

Part of the photovoice focus group was dedicated to receiving residents' feedback on the process and gaining future insights for NeighbourPLAN if it wants to use this method in future neighbourhoods. The group shared the following feedback:

- One month, with a two week-window to take photos, worked well timewise.
- Make a list of things to photograph, but also save a few spots for spontaneous photos.
- Visit places you plan to photograph without a camera first. Sarah notes, "not having the camera with me at first was helpful as I backtracked and remembered how I felt about it".
- Taking photos can lead to awkward interactions; plan for what to say if approached.
- Photo journals were an integral part of the process for most participants, but one felt hindered by it; they provide more support or different ways to reflect.

Disposable cameras were a common frustration. They limited photo quality and quantity. "It was nostalgic, but not in a good way", noted Daniel. Others were disappointed by the overexposed or accidentally taken photos. They acknowledged that not providing cameras would be a barrier for those without resources in their neighbourhood. However, they thought people could bring their cameras, and I could coordinate with other organizations to lend out cameras.

## 6. Discussion

Residents in this photovoice project said seeing the same people, accessibility, green spaces, and cultural events and sites helped them feel connected, while inaccessible design, unaffordable housing, dangerous traffic, and stigmatizing institutions made them feel disconnected. Notably, after photographing their neighbourhood, photovoice participants cited feeling more connected to their neighbourhood. This is a particularly impressive result when comparing it to other NeighbourPLAN engagement activities, about which some of the same residents in the photovoice project later articulated in the evaluation focus groups that those activities did not garner the kind of strong social connectedness they hoped for (McBee 2021). The type of social connectedness residents defined during the evaluation focus groups spoke to more sustained relationships of trust, respect, mutual aid, and support among their neighbours and community. When I asked why connections are so hard to create, some residents from the DTJC felt it was because other residents "don't trust, they're suspicious", or cited a lack of pride in the community and that the participatory engagement strategies that NeighbourPLAN employed felt "too much like work", which led to less long-term participants and less opportunity to develop those relationships (McBee 2021). Rice et al. (2016) write that art "facilitates certain encounters between persons that would not occur otherwise because our communication with one another is so highly regulated" (p. 64), and the literature supports that community artmaking and storytelling can facilitate unique and new connections, especially between stakeholders or individuals whose lives are separated by cultural, social, political, or economic forces (Askins and Pain 2011; Beauregard et al. 2020; Youkhana 2014); however, Lavrinec (2014) highlights how the repetition in playful interactions during large-scale arts-based participatory projects are more helpful in developing emotional connections for a more robust, sustained network.

### 6.1. "Stuff" and Its Potential for Building Connections

Askins and Pain (2011) write that material-based hands-on methods can be valuable instruments for non-verbal, embodied listening. Material objects can also lower the barriers to interactions with strangers from different socioeconomic groups and thus foster more opportunities to connect than traditional research methods engender. In the participatory planning pilot project 'Stewart Street Active Neighbourhood', Nasca (2016) notes the resident steering committee's unanimous positive evaluation of the 3D asset map as a successful relationship-building tool that allowed neighbours to orbit around a physical, interactive object and engage with others. Social Capital Theorist Small (2009) observes that "relationships often arise around a common object of attention", especially if the item engenders cooperative and non-competitive action (p. 25). This photovoice project supports those findings. Residents seamlessly maneuvered through stories of loss, joys, hobbies, and interests when prompted by photos. Residents then physically interacted with the photos, generating further sharing, and finding commonality in resources, strategies, and ways of seeing their community.

The inquiry into the physical materials in participatory research and their effects on power dynamics are under-theorized in PAR literature (Askins and Pain 2011), despite increasing interest in how spatiality affects outcomes of encounters between diverse groups (Askins 2018; Valentine 2008). Askins and Pain (2011) observe the centrality of materials in physical encounters or 'contact zones' to break down patterns of social interactions. In their case study, interracial youth partook in two phases of an afterschool arts program. The first phase was marked with youth doing the art, requiring the use of materials, which

led to positive engagement and a temporary, tenuous collapse of racial divides. In the project's second phase, a local artist designed an art product on behalf of the youth, but in the process, she accidentally made the youth group feel self-conscious by highlighting differences. Without the physical paintbrushes, paints, and paper, fragile bonds fell apart when focusing on otherness. In other words, while it is the act of doing that defines embodied research, the objects we use can serve as helpful catalysts for connection and creating an "axis for emerging citizens' networks" (Lavrinec 2014, p. 59).

Within the photovoice project, the residents moved about and physically interacted with the photos while sharing stories, stimulating further interactions between the group. Through physical images spread across the table, residents could view shared challenges, linking commonalities across photos, thus, situating their personal experiences into broader socio-political realities and biases. This parallels Askins and Pain's (2011) research finding that objects within a contact zone created new knowledge, interactions, and solidarities.

### 6.2. Strengths, Solidarities, and the Dangers of Photovoice

True et al. (2019) describe that they are continually inspired by how their participants note that photovoice increases awareness of "unmet needs, the strengths, and sources of support that exist in their lives" (p. 24). I also witnessed this sentiment among the DTJC group. Residents expressed an awareness of how connected they were, acknowledged that their connections were assets and articulated when those connections were missing. Residents also reflected on the causes for their missing connections, often scaling it up to larger processes of classism or racial discrimination. Strategies for navigating those realities were also shared with examples including engaging and even starting their own food security initiatives, Peterborough Gleans, to help with chronic illnesses, volunteering, and starting a neighbourhood association to have their voices amplified and accompanying their Indigenous friends and family who needed to go to the hospital as they noted the racism they experienced in health care.

Visiting the spectrum of Freire's foundational model of *conscientização*, which details three levels of consciousness in how people perceive their social and political realities: Magical, when participants have internalized beliefs about their inferiority and remain silent and complicit in their marginalization; Naïve, when participants view others as the problem, placing blame on their peers rather than seeing their role and role of the oppressor; and finally, Critical, when participants become aware of the systems of oppression, and begin to reject harmful representations that are used to justify their disenfranchisement; this photovoice project reached 'critical consciousness' and even inspired some "intentions to act", a fourth phase of consciousness more recently identified by Carlson et al. (2006) when residents stated their intention to continue or start up new creative outlets. However, I would not claim this photovoice project "uncovered" any oppression as residents seemed acutely familiar with the forces that kept them marginalized and struggling in poverty; what it did unveil to residents was how connected they were, and how integral they were in creating connections (i.e., assets) in their communities, while also offering some new insights and unity by sharing their experiences with others.

Participatory storytelling methods like photovoice carry the same potential as any participatory method to unintentionally reproduce dynamics of power, privilege, and harm (Murdoch et al. 2016; Taylor and Wei 2020); in particular, depoliticized art or art that focuses on aesthetics over authenticity can unintentionally objectify, oversimplify, and diminish people's identities (Askins and Pain 2011). By leaning into storytelling, photovoice can be one tool to intentionally build social connections and community. Nevertheless, this research approach does not guarantee participant empowerment, especially if engagement is not meaningful throughout the process (Liebenberg 2018), or attention is not adequately paid to action, power dynamics, and broader political processes. Another risk lies around how much participants are asked to disclose, as this requires caution, education, and training because it can make participants vulnerable and even re-traumatized, especially if antagonistic power relations are present (Beauregard et al. 2020). In Taylor and Wei's

(2020) case study of Story Bridge, participants felt optimistic about building trust, but some left feeling vulnerable and raw, saying they had not shared that deeply in decades. This risk reminds me of Tuck's (2009) call for research to focus on desires, which might include stories of hurt, but also hope and creativity, so as not to fetishize loss and ongoing denigrations. In NeighbourPLAN photovoice project, I saw how readily stories of loss, stigma, and hurt came up, but also stories of connection. Residents were encouraged to decide what they wanted to take photos of and develop self-representational narratives via the photo journals, and they should feel autonomy in what they were choosing to share or not. When undertaken thoughtfully, storytelling methods, such as photovoice, can provide a less extractive exploration of lived experiences, encourage creative thinking, active listening, and connection building (Joyce 2018; Taylor and Wei 2020; Wang 1999), and are possible strategies for mitigating methodological harm.

## 7. Conclusions

As a highly regarded community organization in Peterborough, NeighbourPLAN leveraged its network to amplify marginalized experiences and perspectives. NeighbourPLAN's commitment to deep engagement allowed ample opportunity to create a feedback-rich environment. NeighbourPLAN included photovoice into their program design upon my suggestion and I recommended that it include more arts-based activities in future projects. Storytelling and arts can be used in participatory planning to capture nuanced human-urban experiences, reveal surprising connections, translate community knowledge into an accessible medium and encourage discussion and visualization of an issue and the next steps for action. Although no quick fix can be guaranteed when trying to foster and sustain relationships, my perspective is hopeful that when undertaken thoughtfully, arts-based research activities can strengthen social networks and senses of belonging while navigating past social norms (Beauregard et al. 2020). Genuinely facilitating storytelling methods creates opportunities for communities to recognize diversity and commonality, building more trusting relationships (Taylor and Wei 2020). These methods can be used to create more meaningful, deeper connections that many participants seek through community engagement. Photovoice participants in this project jointly reflected on their subjectivities and processed systemic injustices by sharing their stories with one another. This project also helped residents animate and express their community's intangible assets, clarify to the residents what spaces felt welcoming or not, and discuss the reasons behind that. In so doing, the worlds of participants within Peterborough became more visible to NeighbourPLAN and to residents/participants themselves. Complex participatory planning projects such as NeighbourPLAN are massive undertakings that require many skills from many passionate people, but ensuring we provide the time and space for marginalized communities to form and share their stories and lived realities beyond a survey has the potential to generate so much more than data.

**Supplementary Materials:** The final Vision document for the Downtown Jackson Creek Neighbourhood has been published on GreenUP's website. Available online: https://www.greenup.on.ca/wp-content/uploads/2020/04/NP-Vision-DTJC-Web.pdf (accessed on 14 July 2023).

**Funding:** This photovoice project was funded specifically by the Symons Trust Fund for Canadian Studies (700 CAD), and GreenUP which helped cover 50% of the residents' honorariums (300 CAD of 600 CAD). The overall graduate research and evaluation of NeighbourPLAN and subsequent thesis was funded through Mitacs Accelerate (Application Ref. IT14691) and the Social Science and Humanities Research Council (SSHRC) Joseph-Armand Bombardier CGS Award 766-2019-1463.

**Institutional Review Board Statement:** The study was conducted in accordance with the Tri-Council Guidelines (article D.1.6) and approved by the Trent University Research Ethics Board (25743, 5 December 2019).

**Informed Consent Statement:** Informed consent was obtained from all subjects involved in the study.

**Data Availability Statement:** The data presented in this study are openly available in the Graduate thesis by the author, Rosa McBee, in the Trent University Library and Archives found in http://digitalcollections.trentu.ca/collections/trent-university-graduate-thesis-collection (accessed on 28 December 2022), with the title: Building social connections: Evaluating NeighbourPLAN's participatory planning initiative for increased participant connectedness in Peterborough, Ontario.

**Acknowledgments:** I want to acknowledge and pay my respects to the traditional territories and the caretakers of the land where I conducted this photovoice project, Curve Lake First Nation's traditional territory and the Mississauga Anishinaabeg people. Thank you to the professors at Trent University. In particular, my supervisor, Nadine Changfoot, who modelled the kind of researcher I wanted to be. I want to thank NeighbourPLAN staff, in particular, who not only helped fund my research in conjunction with Mitacs Accelerate but who always showed openness and support for my thoughts that fostered an environment of mutual feedback and growth. It was an enormous honour to receive the Social Science and Humanities Research Council research award, and I am also indebted to the other external funding that gave me the freedom to do this research. I want to express my gratitude and thanks to the residents who agreed to take part in this photovoice study, Lori, Connie, Jaclyn, Kelly, Kendra and Rick. I found your photos and reflections a source of tremendous motivation throughout grad school, and I hope I represented your perspectives well.

**Conflicts of Interest:** The author declares no conflict of interest. The funders had no role in the design of the study; in the collection, analyses, or interpretation of data; in the writing of the manuscript; or in the decision to publish the results.

## Notes

[1] A theory of change is an evaluation approach that clarifies the underlying hypothesis of social interventions by creating a diagram that visualizes the causal links between activities and short to long-term anticipated or aspirational changes. ToCs have become a predominant approach among evaluations in Canadian non-governmental, governmental, and higher education circles conducting CBR or PAR as a framework to compare the intended versus the actual outcomes (Funnell and Rogers 2011).

[2] SHOwED is an acronym for a series of questions to guide photo journaling for each photograph to identify the strengths and challenges and prompt the photographer to think about how the situation might be improved.

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
