# Peer review of "“It Really Put a Change on Me”: Visualizing (Dis)connections within a Photovoice Project in Peterborough/Nogojiwanong, Ontario"

_socsci, doi:10.3390/socsci12090488_

Round 1

Reviewer 1 Report

Please see attached report

Author Response

Review for Social Sciences Journal. "It really put a change on me": Visualizing (dis)connections 2 within a photovoice project in [City Name], Ontario

Thank you for the opportunity to review "It really put a change on me": Visualizing (dis)connections 2 within a photovoice project in [City Name], Ontario. This is an interesting paper that serves to describe the experiences of individuals living in the DTJC community along with a critical analysis of the Photovoice methodology within the project. The narratives included in the findings were useful in articulating the experiences of the participants and illustrating the various issues encountered by the community. I appreciate the author’s critical assessment of PAR and their self-reflection of the entire process, especially of the methodology.

I found a critical analysis of Photovoice seems to be lacking both in the background section and in the discussion. Given that the intent of the article is to evaluate the photovoice method with this group, the paper would be strengthened by identifying of the 3 main goals of photovoice as outlined by Wang and Burris 1997 should be included. I also recommend inclusion and reflection of Liebenberg, L. (2022). Photovoice and Being Intentional About Empowerment. Health Promotion Practice, 23(2), 267–273. https://doi.org/10.1177/15248399211062902 and Liebenberg, L. (2018). Thinking critically about Photovoice: Achieving empowerment and social change. International Journal of Qualitative Methods, 17(1), 1609406918757631. https://doi.org/10.1177/1609406918757631 both in the background section and as part of the discussion on the importance of empowerment in photovoice.

Thank you very much for these concise and constructive suggestions. I added Wang and Burris' three main photovoice goals to the methodology section's first paragraph (see lines 214-219). Thank you as well for recommending Liebenberg's articles; I am glad to be introduced and wish I had read this earlier! I incorporated a few arguments and points that made me reflect more on the research design and discussion (see lines 291, 300-303, 593-594). In my thesis, I wrote in my literature review about ‘empowerment’ in PAR and why I tend to avoid using the term except in a few cases. In short, I find the critique from Cahill (2007) convincing; namely, the term 'empowerment' has a very paternalistic, assimilationist history and is contradictory to PAR's premise that everyone already, inherently, has power. However, Wallerstein and Bernstein's definition of the "power to act with others to effect change," specifically in shaping social policy and social change, as cited in Liebenberg's article, is a more agreeable definition. I must consult more literature on the topic for future research. I tried to include what I could while treading lightly around using the argument for "empowerment," but I think what I added regarding Freire’s critical consciousness-raising (see lines 577-586) help to address this idea without using the word “empowerment.”

There is also no inclusion of Photovoice’s theoretical foundations to Freire’s critical consciousness work and his work on empowerment. In Line 302-303 in the findings, the author states “residents' reflections quickly revealed broader power deficits”. It would be useful in the discussion to return to this concept as part of the critical dialogue and consciousness that is important in Photovoice. I would further recommend Carlson, E. D., Engebretson, J., & Chamberlain, R. M. (2006). Photovoice as a Social Process of Critical Consciousness. Qualitative Health Research, 16(6), 836–852. https://doi.org/10.1177/1049732306287525

I have added a citation to Freire in the Introduction (see lines 42-45) and to the discussion section (see lines 577-586) regarding his take on critical consciousness raising. I also included Carlson et al. 2006 in the same location in the Discussion notes (see lines 580-581) regarding their contribution of “intention to act” as a part of critical consciousness raising. Thank you for this suggestion.

Final note: Lines 76-78 – are they included in error?

Yes! Apologies. I removed those lines.

Line 211 – It would be helpful to have a footnote explaining SHOwED.

I added a footnote (see page 6 of 19)

The long footnotes on page 2 were distracting. I would encourage them to be more succinct.

 Thanks for this feedback; I made the wording a bit more concise in a few places, and I cut out the part about Tuck (2009) and added those lines to the discussion (see lines 601-603). I also split up the footnote about “social connectedness” vs “social capital” and added a footnote at the bottom of page 1, so page 2 footnotes are shorter.

Reviewer 2 Report

Thank you for the invitation to review the manuscript   "It really put a change on me": Visualizing (dis)connections 2 within a photovoice project in [City Name], Ontario.” I was delighted to see a paper focused on photovoice as a method for community engagement in planning.

I recommend this manuscript for publication with revisions.  There are a lot of ideas in the paper, perhaps too many. The arguments and writing need to be clarified.

My primary feedback is that the paper needs to be reorganized in a way that the paper develops themes introduced in the introduction. The introduction to the paper is strong and lays out a few lines of inquiry. First, how photovoice might contribute to city planning (as stated in the introduction to the paper). Second, the author suggests that although the photovoice process focused on reflecting upon disconnections, in fact through the photovoice process the community came to feel more connected. This is interesting and I would love to learn more!

In the abstract the authors identify that

1.      The findings from this project determine that green spaces, non-judgmental institutions, accessible amenities, safe housing, and well-maintained streets were critical for resident researchers' feelings of connectedness.

2.     Additionally, the findings also conclude photovoice generated what Cahill (2007) describes as 'developing new subjectivities,' as residents reported feeling more connected to their community after taking photos

The two lines of inquiry seem quite fruitful, but because there is quite a bit of discussion of other aspects of the research, the writing/argument gets confusing to follow at times (there is a lot of details about the process and/or critiques of the literature that, while interesting, might belong in another paper)

The literature review on intentional connection building in PAR is interesting. It would be meaningful to bring the frame of relationality into the findings discussion in a way that holds the complexity and depth of analysis. At the moment the article feels a bit descriptive and not theoretically engaged.

As a participatory process, one of photovoice’s strengths comes from the reflection, discussion, consensus and meaning-making through the process, with a particular audience in mind. It would be good to get a sense of this in the process, and how this contributes to shifting subjectivities – and how this might be useful to planners too (along the lines of inquiry introduced in the introduction, discussed above).  This connects with the discussion on page 13 of materiality, which I think could be connected with a sense of audience and also how through collective analysis they are developing new subjectivities that are relational and connected with community.

I think there is a lot to work with, but the article needs a judicious edit as well as an elaboration . 

I appreciate this definition upfront in the article:

 Photovoice is a qualitative, arts-based participatory research method that realizes the spirit of PAR as it invites participants to identify, collect, and analyze their world, realities, and experiences by taking photographs during a set window of time. The photographs encourage creative reflection on how to visually represent assets and challenges in a community, which is accompanied by critical dialogue through focus groups, or public exhibitions around selected images (Hergenrather et al. 2009; Joyce 2018; Wang and Burris 1997). This article describes a photovoice project run within the participatory planning project …

What needs to be developed is an understanding of the critical dialogue and meaning making and how this might inform city planning and the development of new subjectivities.

I also love that although the focus was to capture disconnections, through the process, “of reflection, documentation and photo-storytelling, residents shifted their experience of their community to feel more connected and clarified the spaces and places that offered safety and those that did not…. . Even ephemerally, the photovoice project made more visible for participants their worlds within [City Name] for one  another. ” 

But then this line of inquiry doesn’t get developed enough.  

I worried that the authors might be trying to follow an article structure/ format that is usually used for quantitative research, with a findings, limitations and discussion section, and that this format truncated the depth of the analysis. The discussion section and the section on shifting subjectivities need further elaboration in conversation with the literature. The conclusion is underdeveloped.

In sum, I suggest reorganizing the paper to develop the themes in a way that builds/contributes to how photovoice might be a generative method for planners to use in community engagement. For example, can the author say more about how “The findings from this project determine that green spaces, non judgmental institutions, accessible amenities, safe housing, and well-maintained streets were critical for resident researchers' feelings of connectedness” might be useful to planners? And why and how planners might work with this method. I think this could be a significant contribution if reworked.

See notes below. I hope this is helpful!

Notes :

The footnotes are too long. I appreciate so many of the ideas contained within them. Suggest integrating the material into the narrative text.

Page 3, line 76 – this sentence is awkwardly phrased -is it an editorial comment  left in?

This section may be divided by subheadings. It should provide a concise and precise 76

description of the experimental results, their interpretation, as well as the experimental 77

conclusions that can be drawn.

p. 3  Suggest rephrasing – the word immeasurable connotes positivism, but there is qualitative research no? sentence is also long ..

“The interdisciplinary research and evidence of social connectedness as an important fac- 97

tor for health and wellness is immeasurable, and factors such as individual and commu- 98

nity resilience, diagnoses and recovery all improve with the presence of meaningful con- 99

nections (Comes 2016; Griffiths et al. 2007; Hari 2018; Taylor and Wei 2020; Umberson 100

and Montez 2010).

p. 3 why is social capital capitalized?

Suggest framing this as a critique of positivism and the bias of funding agencies toward quantitative research. There is a body of research supporting qualitative and participatory research in evaluation.

p. 3 However, relationships and connections are difficult to anticipate, explain, and evaluate, 121

particularly when a funding agency asks for tangible outcomes and impact evaluation 122

(Hardy et al. 2018; Levkoe and Kepkiewicz 2020).

Suggest rephrasing or deleting “The only paper I found”

p. 4 Hardy et al.’s (2018) case study of  Hermosas Vidas is the only paper I found that tried to quantify the number of lasting relationships built through the participatory health research project

p. 4 too many theories that have to be explained : Meshworks theory, Ripple theory etc…

p. 4 p 151 Include citations:

 Literature about participatory research discusses connectivity and quintessential 151

social networks but does not often explain the decisions on how to develop them.

p. 4 Despite the claim in participatory literature that building relationships within partic- 161

ipatory projects is essential and leads to more action (Janzen et al. 2016), there is a gap in 162

the participatory literature on how connections and relationships are defined and how 163

they are/were formed. 

Suggest doing more extensive review of literature beyond quantitative data to cite work on care. Thinking about Duncan-Andrade’s work on cariño. Thinking about feminist PAR work on relationships as central to the inquiry and analysis.

p. 4 a theory of change is not just a diagram but an evaluation approach/model

 A theory of change is a diagram that clarifies the underlying hypothesis of social interventions by drawing causal links between activities and short to long-term anticipated or aspirational changes

p. 5  what about race/ethnicity?  And how many people ?

p. 6 first paragraph . confusing. Suggest deleting or rephrasing. .

6 Firstly, I selected the topic of connectedness from the Theory

of Change, created by [Non-profit Urban Sustainability Organization in City] before

they entered the neighbourhoods.

But I thought you stated earlier that residents identified connection as important to them on an earlier survey, no?  This, then, would make sense as a frame to work with. I am not sure, however, why it was shifted to Disconnections.

p. 6 The final phase of photovoice typically showcases photos to inform policymakers

or the community (Wang 1999; Joyce 2018).

my understanding is that as a PAR method the strength comes from the reflection/ discussion and consensus and meaning making through the process..

p. 6 Since I did not consider this when designing the photovoice project, residents thought of

an audience more abstractly, leading to many themes in their photos. This abstraction

can be beneficial or detrimental, depending on the goal of the photovoice project.

The fact that the residents thought of the audience, as opposed to you, is potentially a quite interesting finding and arguably more participatory. I would love to hear more about this. ..

p. 7 cite Torre & Ayala’s work on contact zones

p. 7  my larger thesis ? how is this relevant?

282  - I did not use a disability arts framework for this photovoice project; I did not specifically

recruit on the topic of disability, and my larger thesis project focused on a class analysis.

p.6- 7 I find the limitations section confusing. Suggest deleting.

p. 9 reading raised questions for me as to how participatory the method is. would love to get a sense ..

would love for more background on the process itself/ the community / the city and how these photos were used ..

p. 12 5.6 resident feedback section, including the paragraph on disposable cameras feels too granular for this type of paper which is not solely focused on methods.  Suggest deleting . Instead focus on the content of their research/photovoice/ and the relationships and connections.

Page 12 – 13 – I wish for more developed discussion of connections and their findings.  The

residents later articulated this further during the evaluation focus groups as a deeper, 466

more sustained relationship of trust, respect, mutual aid, and support.”

This feels significant. Would love to hear more.

This feels connected with the development of new subjectivities.

Page 13 – interesting discussion about “stuff” and materiality as a way of building connection through participatory arts/engagement/ action. While the author hasn’t found a source “ that explicitly considers the phenomenological theory of photovoice” as they state – there may be parallels from Askins and Pain.

I would love to hear more about this and suggest the author unpacks this as this seems like one of their primary contributions and connected with the development of new subjectivities.

The “stuff” and materiality of PAR is, I believe, connected with the “action” phase of PAR and is part of the ontology and epistemology of participatory research.  Action doesn’t necessarily have to be directed outwards, although outsiders may be invited to listen in. My understanding is that a critical part of the photovoice process is collective storytelling through image-making and this in itself is an action that creates a space for participants to reflect upon their lives with others, and in so doing move from personal experience to social theorizing.  

Page 14, line 531.  Furthermore, I bring up Askins and Pain because they so candidly critique their research design and making the mistakes in research visible is such a 532

crucial and valuable practice.

Suggest deleting this. While this analysis is important, I worry that this then loses the argument that the author was building about materiality, which could be further developed and connected with new subjectivities.

The conclusion needs to be further developed. At the moment it feels like a summary that sacrifices a depth of analysis.

Author Response

Please see attachment, thank you. 
